# Response to hepatitis B virus vaccination in individuals with chronic hepatitis C virus infection

**Ashraf A. Ashhab**[1], **Holly Rodin**[2], **Marilia Campos**[3], **Ahmad Abu-Sulb**[4], **Jane A. Hall**[2], **Jesse Powell**[3], **Jose D. Debes**[3,5]*

1 Division of Digestive and Liver Diseases, Cedars-Sinai Medical Center, Los Angeles, California, United States of America, 2 Analytic Center of Excellence, Hennepin County Medical Center, Minneapolis, Minnesota, United States of America, 3 Department of Medicine, Gastroenterology and Hepatology, Hennepin County Medical Center, Minneapolis, Minnesota, United States of America, 4 Division of Pediatrics, Legacy Community Health, Houston, Texas, United States of America, 5 Department of Medicine, University of Minnesota, Minneapolis, Minnesota, United States of America

* debes003@umn.edu

## Abstract

### Background

Previous reports show conflicting results regarding hepatitis B virus (HBV) vaccine efficacy in Hepatitis C virus (HCV)-infected individuals.

### Aims

To evaluate HBV-vaccine response and identify possible factors that may contribute to lower vaccine efficacy in patients infected with HCV.

### Methods

We retrospectively evaluated all patients with chronic HCV infection at Hennepin County Medical Center, in Minneapolis, Minnesota, between 2002 and 2018. We addressed laboratory, liver-related, virus-related as well as vaccine-related variables, and their association to HBV vaccine response. Differences were tested using either a Chi-squared test or a T test to compare means between the two populations. Multivariate regression was modeled as a logistic regression.

### Results

1506 patients were evaluated, of which 525 received appropriate HBV vaccination and were assessed for response. Among those, 79% were vaccine responders and 21% were non-responders. On multivariate analysis, cirrhosis was associated with lower response to the vaccine (OR 0.6, CI 0.44–0.94). We found no significant differences for vaccine response in relation to smoking (87% vs 86%), IV drug abuse (74% vs 72%), Diabetes Mellitus (26% vs 22%) being on hemodialysis (2% vs.5%), or virus related variables.

**Data Availability Statement:** All relevant data are within the manuscript and its Supporting Information files.

**Funding:** JD is supported by the Robert Wood Johnson Foundation, Harold Amos Medical Faculty Development Program(AMFDP) and National Institute of Health (NIH) -NCI R21 CA215883-01A1. URLs: Robert Wood Johnson Foundation: www.rwjf.org Harold Amos Medical Faculty Development Program: www.amfdp.org/ National institute of health: www.nih.gov The funders had no role in study design, data collection and analysis, decision to publish, or preparation of the manuscript.

**Competing interests:** The authors have declared that no competing interests exist.

## Conclusion

HCV infection seems to impair HBV vaccine response, with cirrhosis being the only identifiable risk factor for hypo-responsiveness among studied clinical and virus-related variables.

## Introduction

Hepatitis B virus (HBV) and hepatitis C virus (HCV) infections are the most common causes of chronic liver disease leading to cirrhosis and hepatocellular carcinoma worldwide. The world health organization estimates that in 2015, 257 million people were living with chronic HBV infection, and 71 million were living with chronic HCV [1,2].

Dual infection with HBV and HCV is not uncommon, as both viruses share some common paths of transmission and risk factors, including IV drug use, hemodialysis, and frequent blood transfusions such as in hemophilia [3].

Indeed, more than 25% of HCV-positive patients in the United States had positive markers for hepatitis B exposure, a proportion nearly six times that in the HCV-negative group [4]. Higher rates of cirrhosis and increased severity of liver disease have been reported with co-infection compared to either HBV or HCV mono-infection [5,6]. Additionally, HBV re-activation has been reported in HCV and HBV co-infected individuals receiving direct-acting antiviral (DAA) treatment, including individuals with either current or previous HBV infection [7,8]. Despite the availability of new DAA treatment for HCV and the realistic potential of HCV elimination, many HCV-treated patients continue to exhibit high risk behavior after treatment, putting them at risk for re-infection [9].

As such, HBV vaccine is recommended as the primary means to prevent HBV super-infection and its associated increase in morbidity and mortality in HCV-infected subjects [10].

In this regard, several reports suggest a similar HBV vaccine response rate among HCV infected individuals [11,12], while others suggest that this population mounts a poorer vaccine response [13,14]. This hypo-responsiveness is thought to be related to alterations in the cellular and humoral arms of the immune response secondary to HCV infection [15,16]. The impact of viral related variables such as viral load and viral genotype are controversial [13,17].

In light of the current availability of new DAAs which have effectively turned HCV infection into a curable disease, with well over 90% sustained virological response in most genotypes, attention should be focused towards HCV-related complications and indirect effects, such as poor vaccination responsiveness [18]. In this study we investigated the effect of HCV infection on HBV vaccination response, and the impact of clinical variables, life style, and virus related variables on vaccination response.

## Methods

### Study design and patients

This is a retrospective observational cohort study conducted through chart review of patients. Characteristics, clinical and virus-related variables of individuals infected with HCV who received HBV vaccination in Hennepin Healthcare, a county hospital in Midwestern US, between the years of 2002–2018 were reviewed.

For this study, 1506 electronic medical records (EMR) of individuals diagnosed with HCV (all available) were reviewed. The institutional review board of Hennepin healthcare approved the study. The data were analyzed anonymously therefore consent was not obtained from

individual patients. The primary objective was to assess the response rate of patients with HCV to HBV vaccination, and the impact of clinical and virus related variables on response rate.

Clinical information of the patients were collected from EMR by the participating investigators. The data extracted included patients' age at the time of Hepatitis B Surface Antibody (HBsAb) test, number of HBV vaccine doses received, potential risk factors (*ie*, smoking, IV drug use and alcohol use), presence of cirrhosis, Diabetes mellitus (DM), End Stage Renal Disease (ESRD) requiring dialysis, HCV viral load, and HCV genotypes. Patients with a positive hepatitis B core antibody were excluded from the study.

## Viral definitions

Patients were considered to have chronic HCV infection if they had a positive polymerase chain reaction assay (PCR) for HCV RNA at the time of receipt of HBV vaccination, and at the time of the HBsAb test check.

A positive HBsAb test was defined as having a titer of 12 mIU/mL or above. Patients were considered to be vaccine responsive if they developed a positive HBsAb test following at least one dose of HBV vaccine. Vaccine non-responders were defined as patients who did not develop a HBsAb following at least 3 vaccine doses. Complete vaccination was performed by injection of 20 μg recombinant HBsAg into the deltoid muscle at months 0, 1 and 6. Patients involved in the study received any of the following brands for HBV vaccination; Glaxo smith kline, Glaxo wellcome, Merck, Smith Beecham, Sanofi, Aventis, Abbot, and JHP. The majority of patients with known dates of the first vaccination dose received the first dose prior to HCV diagnosis (69%), however, most of the patients included in the study expressed a remote history of risk factors for HCV infection, suggesting early acquisition of HCV infection. Patients positive for Hepatitis B core antibody were removed from the analysis.

## Clinical definitions

Patients were considered to be cirrhotic at the time of the diagnosis if: a) they had a persistent International normalized ratio (INR) level of $> 1.2$, persistent high levels of total bilirubin ($> 1.2$ mg\dL) in addition to low platelets ($< 150,000$\mm) and radiographic cirrhotic morphology of the liver; b) cirrhotic liver morphology on imaging in addition to an INR of $> 1.2$, c) A combination of low platelet levels with high total bilirubin levels and an INR of $> 1.2$ at any time, d) liver biopsy or fibroscan indicating cirrhosis, or e) they had clinical stigmata of end stage liver disease (ESLD) [19]. Patients were considered to have ESRD if they had a glomerular filtration rate (GFR) of less than 15 mL/min/1.73 m$^2$ and were receiving hemodialysis [20]. Patients were placed into a responsive group and a non-responsive group.

## Statistics

Statistical analyses were performed using SAS version 9.4. Differences between continuous variables were evaluated at the means using a T-Test. Categorical variables were evaluated using a Chi-Square test. The multivariate regression was modeled as a logistic regression.

# Results

## Demographics

We reviewed information of 1506 patients with chronic HCV infection. Of those, 85% percent (n = 1276) received HBV vaccination, 50% (n = 646) of which were checked with HBsAb test afterwards.

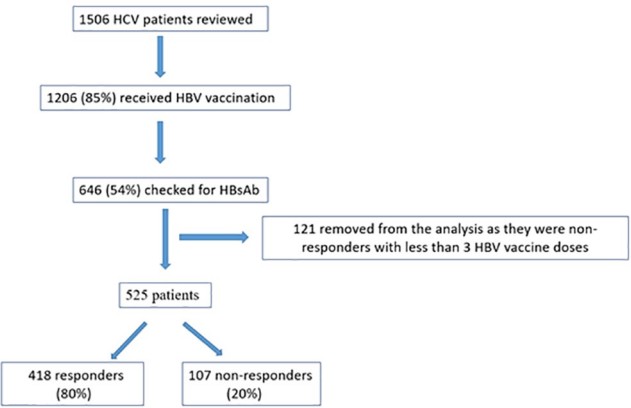

**Fig 1. Vaccine responders and non-responders involved in the final analysis.**

When comparing the patients who underwent HBsAb testing post vaccination to those who did not, both groups showed male predominance (70% vs 63%, respectively) and the group with post vaccination testing was slightly older, with mean ages of 54 (IQR 53–54) and 48 (IQR 47–50), p = 0.0001, in each groups respectively. Additionally, those who were not tested for HBsAb had lower rates of cirrhosis (14% vs 25%, p = 0.01).

Out of the 646 patients who were evaluated with HBsAb testing, 121 patients were excluded from the analysis as they were non-responders but had received less than 3 HBV vaccine doses.

Among the remaining 525 patients, 80% (n = 418) developed HBsAb after vaccination with any number of HBV vaccine doses (vaccine responders) whereas 20% (n = 107) did not develop the antibody, following 3 or more HBV vaccine doses (vaccine non-responders) (Fig 1). Out of all the patients who received 3 or more doses in both groups, only 50% developed HBsAb.

When characterizing patients by the number of the HBV vaccines administered, 23% received 1 vaccine dose, 25% received 2 doses, 37% received 3, and 15% received over 3 HBV vaccines. Among the patients who received 2 doses, the response rate was 17%, compared to 42% in patients that received 3 doses (Table 1). The average number of vaccine doses in responders was 2.6, compared to 3.6 in non-responders (p<0.001) (Table 2).

The demographics of the non-responder group were similar to the responders, with males comprising 65% and 69% of the cohort, respectively (p = 0.4). The median age at the time of the HBsAb test check was 57 years in non-responders, compared to 54 years in responders (IQR: 49–63 and 47–59, for non-responders and responders, respectively).

**Table 1. HCV patients' response following various number of HBV vaccine doses.**

| Number of HBV vaccine doses | Proportion of HCV patients which were vaccinated (n = 413) | Proportion of responders (n = 185) | Proportion of non-responders to any number of vaccines (n = 228) |
| --- | --- | --- | --- |
| 1 dose | 23% (n = 95) | 25% (n = 46) | 21% (n = 49) |
| 2 doses | 25% (n = 103) | 17% (n = 31) | 32% (n = 72) |
| 3 doses | 37% (n = 153) | 42% (n = 79) | 32% (n = 74) |
| ≥4 doses | 15% (n = 62) | 16% (n = 29) | 15% (n = 33) |

**Table 2. Demographics, clinical and virus related-variables for responders with any number of doses vs non-responders.**

|  | Responders (n = 418) | Non-responders n = 107 | P value |
|---|---|---|---|
| **Average age at the time of HBsAB test** | 52 | 51 | 0.6 |
| **Males (%)** | 69% | 65% | 0.4 |
| **Females (%)** | 31% | 36% | 0.4 |
| **Average number of doses** | 2.6 | 3.6 | <0.001 |
| **Presence of cirrhosis (%)** | 24% | 29% | 0.4 |
| **Alcohol use (%)** | 79% | 75% | 0.4 |
| **Tobacco use (%)** | 87% | 86% | 0.8 |
| **IV drug use (%)** | 74% | 72% | 0.6 |
| **Diabetes (%)** | 26% | 22% | 0.4 |
| **Dialysis (%)** | 2% | 5% | 0.1 |
| **Average HCV-RNA at HBsAb test check** | 3246751 | 3586957 | 0.4 |
| **Genotype 1a** |  |  |  |
| **Genotype 1b** |  |  |  |
| **Total genotype 1 (%)** | 80% | 79% | 0.9 |
| Genoype 3 | 10% | 10% |  |
| Other genotypes | 10% | 10% |  |

## Clinical variables

When comparing the clinical variables among the responders and the non-responders groups, we found no differences between the rates of alcohol use (79% vs 75%), IVDU (74% vs 72%) or Diabetes Mellitus (26% vs 22%), respectively. The proportion of smokers in both groups was similar (87% and 86% respectively). Only 5% of non-responders had end-stage renal disease and were on dialysis, compared to 2% of the responders (p = 0.1).

Notably, when assessing for the presence of cirrhosis, the rate of non-responders with cirrhosis was higher than that in the responders group (29% vs 24%), but the difference was not statistically significant (p = 0.4), (Table 2). However, on multivariate analyses that included age, gender, cirrhosis, alcohol abuse, and DM, only patients with liver cirrhosis were less likely to be reactive (OR 0.66, CI 0.44–0.94). When evaluating all cirrhotic patients who received 3 or more doses in the study, we found a response rate of 77% (n = 32/139).

A total of 393 HCV patients did not have cirrhosis in our cohort. Of those, 81% (n = 317) were responders (following any number of vaccine doses), whereas 19% (n = 76) were not (following at least 3 vaccine doses). When non-cirrhotics who received at least 3 vaccine doses in both arms were assessed, the response rate was found to be 51% (79/155).

## Virus-related variables

The average HCV viral load at the time of HBsAb test check among both of the groups was similar (3.5 x10$^6$ vs 3.2x10$^6$ IU/mL, for responders and non-responders respectively, p = 0.4). Both of the subgroups also had similar proportions of different HCV genotypes (Table 2).

Notably, all the above clinical and virus-related variables were also compared between responders who received 3 or more vaccine doses and non-responders, with the comparison yielding similar results (Table 3).

## Multivariate analysis

On multivariate analysis that included age, gender, cirrhosis, alcohol abuse, and DM, patients with liver cirrhosis were less likely to be re-active (OR 0.6, CI 0.44–0.94). Gender, age, alcohol

**Table 3. Demographics, clinical and virus related-variables for responders who received ≥3 doses vs non-responders.**

| Variable | Responders (N = 107) | Non-responders (N = 107) | P value |
|---|---|---|---|
| Average age at the time of HBsAb test | 51 | 51 | 1 |
| Males (%) | 67% | 65% | 0.7 |
| Females (%) | 33% | 35% | 0.7 |
| Average number of doses in those with ≧3 doses | 3.5 | 3.6 | 0.5 |
| Presence of cirrhosis (%) | 26% | 29% | 0.6 |
| Alcohol use (%) | 94% | 75% | <0.0001 |
| Tobacco use (%) | 91% | 86% | 0.3 |
| IV drug use (%) | 72% | 72% | 0.6 |
| Diabetes (%) | 26% | 22% | 0.5 |
| Dialysis (%) | 2% | 5% | 0.3 |
| Average HCV-RNA at HBsAb check | 3220061 | 3586957 | 0.5 |
| Genotype 1a | | | |
| Genotype 1b | | | |
| Total genotype 1 | 82 | 79 | 0.26 |
| Genotype 3 | 13 | 11 | |
| Other genotypes | 5 | 10 | |

abuse and DM did not show a significant correlation with HBV vaccine reactivity. These variables have been suggested before to correlate with incomplete response to vaccination [21,22].

## Discussion

To our knowledge, this is one of the largest studies to date to assess HBV vaccine responsiveness in HCV infected individuals, including both clinical and virus-related variables, and various numbers of vaccine doses. We found a significantly lower response to HBV vaccination in HCV-infected individuals, with an overall response rate of 79%, and a response rate of 50% when the cohort of those who received 3 or more vaccine doses was assessed. This is significantly lower when compared to the response rate of 90% to 98% reported in the general population [11,23]. Previous reports have found similar results, with sub-optimal to significantly lower response rates in HCV patients [24,25]. A rapid decline in the proportion of patients with protective levels of HBsAb over 1 year has also been reported in this population [26].

Interestingly, a number of studies found the HBV vaccine response rate to be similar regardless of HCV status [27,28]. In a recent meta-analysis of 11 studies involving 704 HCV patients, Liu et al concluded that chronic HCV infection can decrease the immune response to a standard schedule of hepatitis B vaccination [29]. Our study validates this finding with the advantage of having a similar number of patients within one study.

The cause for the lower vaccine response in HCV-infected individuals appears to be multifactorial. HCV patients who are HBV vaccine non-responders were found to have up-regulation of PD-1, a negative immune-modulator that correlates with diminished T cell activation in response to both general and virus-specific stimulation [15]. In a murine model, HCV core has been found to strongly inhibit the cytotoxic immune response [30,31]. Lastly, impairment in the humoral arm of the immune system has been noted in HCV patients and implicated in their poor HBV vaccine response [16]. Interestingly, some reports also suggest that HBV vaccine non-responders exhibit poor responses to tetanus toxoid and Candida as well [32].

Overall, the presence of these immune changes in HCV-infected individuals likely plays a large role in vaccine hypo-responsiveness, and the extent to which immune response is decreased could be a factor related to the discrepancy between some of the studies. Genetic

variation could be also related to the discrepancy in vaccine response, with genetically deter-mined low-responsiveness to HBsAg being reported in carriers of different HLA types such as B8, B44, DR3, DR7, and DQ2, however, this is less likely to be a major contributor to the response discrepancy among HCV patients [33,34].

In the vaccine non-responder population of our study there was a trend of no-response associated to the presence of cirrhosis, which was not significant on univariate analysis, but became evident on multivariate analysis. It has been reported that cirrhosis is associated with immune dysfunction involving both B-memory-cells, and T-cells, including helper T-cells, both of which are essential in mounting immunity to HBV vaccine [16,35]. Indeed, cirrhosis has been implicated in poor responsiveness to HBV vaccine in previous reports [36,37]. Simi-lar to other studies, we found that clinical variables including age, gender, alcohol use, IV drug use and smoking were not significantly different between responders and non-responders [17,23]. DM was found to be a poor predictor for HBV vaccine response in patients with Chronic kidney disease (CKD) and was associated with lower response to both HAV and HBV vaccines when co-present with both HCV and fibrosis [38,39]. However, our study found simi-lar rates of Diabetes Mellitus (DM) among responders and non-responders.

Virus-related variables such as HCV genotype and viral load have been shown to play a role in HBV vaccine responsiveness. One study showed a worse response to HBV vaccine in patients who had been infected by genotype-1 of HCV as compared to those infected by geno-type 2 or 3, and some manuscripts have reported an inverse correlation between increased HCV viral load and response to HBV vaccine [18,22]. However, our study found no negative impact by HCV viral load or genotype on response to vaccination. This finding may be further suggested by reports noting no increase in HBV vaccine sero-conversion following HCV treat-ment with achievement of sustained virological response [26].

There is no current consensus regarding a differential HBV vaccination regimen in HCV patients. In our study, some vaccine non-responders seemed to develop sero-conversion with additional dose administration. Other vaccination regimens have been previously suggested in the literature. Monkari et al. achieved a higher response rate with double dose HBV vaccine in HCV patients compared to the standard dose, with 100% of their patients responding to 40 µgs administered at the standard 0, 1, 6 month regimen [40]. In the current age of DAAs however, re-vaccination for HBV post DAA treatment seems a sensible option to achieve max-imum success.

There are several limitations to our study. Firstly, it is retrospective in nature. We did not characterize cirrhotic patients by MELD or Child-Pugh scores or diabetics by glycemic con-trol, limiting our assessment of the impact in advanced stages of these disorders. Additionally, while a large proportion of our patients received over 3 vaccines to achieve sero-conversion, we did not assess the timeline during which extra doses were administered, and subsequently any possible variation in response rates based on early vs late administration.

In conclusion, we found that HCV infection significantly impairs HBV vaccine response, with liver cirrhosis being the main risk factor for vaccine hypo-responsiveness. We found no significant association of other clinical variables or virus-related variables with vaccine response rates. Prospective studies with HBV vaccination following DAA treatment are warranted.

## Supporting information

**S1 File.**
(XLSX)

## Author Contributions

**Conceptualization:** Ashraf A. Ashhab, Jose D. Debes.

**Data curation:** Holly Rodin, Jane A. Hall, Jesse Powell, Jose D. Debes.

**Formal analysis:** Holly Rodin, Jane A. Hall.

**Supervision:** Jose D. Debes.

**Writing – original draft:** Ashraf A. Ashhab, Marilia Campos, Jesse Powell, Jose D. Debes.

**Writing – review & editing:** Ashraf A. Ashhab, Ahmad Abu-Sulb, Jose D. Debes.

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
