## [Decision Letter · Decision Letter 0]

9 Apr 2020

PONE-D-20-06696

Response to hepatitis B virus vaccination in individuals with chronic hepatitis C virus infection

PLOS ONE

Dear Dr. Ashhab,

Thank you for submitting your manuscript to PLOS ONE. After careful consideration, we feel that it has merit but does not fully meet PLOS ONE’s publication criteria as it currently stands. Therefore, we invite you to submit a revised version of the manuscript that addresses the points raised during the review process.

We would appreciate receiving your revised manuscript by May 24 2020 11:59PM. To enhance the reproducibility of your results, we recommend that if applicable you deposit your laboratory protocols in protocols.io, where a protocol can be assigned its own identifier (DOI) such that it can be cited independently in the future. For instructions see: http://journals.plos.org/plosone/s/submission-guidelines#loc-laboratory-protocols

We look forward to receiving your revised manuscript.

Kind regards,

Jason Blackard, PhD

Academic Editor

PLOS ONE

Additional Editor Comments (if provided):

This is a retrospective analysis of HBV vaccine efficacy in HCV co-infected individuals in the US.

A relatively conservative estimate of HBV vaccine protectiveness was used.  What if that cutoff were increased to 100?

Were the sociodemographic or clinical characteristics of those that were checked for anti-HBs different from those that were not checked? (regardless of the findings)

What was the HIV status of the individuals in the responder versus non-responder groups?

How were alcohol and intravenous drug use defined?  Self-report?  Ever / never?

Journal Requirements:

2. In the ethics statement in the manuscript, please provide additional information about the patient records used in your retrospective study. Specifically, please ensure that you have discussed whether all data were fully anonymized before you accessed them and/or whether the IRB or ethics committee waived the requirement for informed consent. If patients provided informed written consent to have data from their medical records used in research, please include this information.

Reviewers' comments:

Reviewer's Responses to Questions

**Comments to the Author**

1. Is the manuscript technically sound, and do the data support the conclusions?

Reviewer #1: Partly

Reviewer #2: Partly

2. Has the statistical analysis been performed appropriately and rigorously? 

Reviewer #1: No

Reviewer #2: No

3. Have the authors made all data underlying the findings in their manuscript fully available?

Reviewer #1: Yes

Reviewer #2: No

4. Is the manuscript presented in an intelligible fashion and written in standard English?

Reviewer #1: Yes

Reviewer #2: No

5. Review Comments to the Author

Reviewer #1: This manuscript describes the results of a retrospective evaluation of HB vaccine response in a large group of HCV infected patients in Hennepin County, MN. The authors conclude that HCV infection impairs HBV vaccine response rates.

Specific Comments

1. Page 5: The sentence regarding vaccination with Euvax B is very confusing. This vaccine is not licensed in the U.S. and the next sentence describes other vaccine products used.

2. I disagree with defining vaccine non-responders as those that received at least 1 dose. While this may be the definition in an intent to treat trial, we know from prospective registration trials that the rate of response from those who received only 1 dose is quite low and 2 doses is suboptimal. In the original registration trials from Szumness, the post-vaccine response (defined as >10 mIU/ml) was around 30% and this went up to around 70% after the second dose. In prospective trials the testing was done at short time intervals while your testing was randomly done, also decreasing apparent outcomes. At the minimum your study should focus on those who completed a full series.

3. Page 6: Your definitions of cirrhosis are very non-specific. Why not apply FIB-4 which is imperfect but is at least standardized in HCV positive patients and call cirrhosis those with FIB-4 >3.25? Use of mildly increased bilirubin in the definition allows inclusion of those with Gilbert's gene polymorphisms.

4. Page 6: Only a subset of patients had testing for surface antibody after vaccination. How did these differ from the non-tested population?

5. Page 8: Comments about genetic variation not germain to your results. Indeed, with general population rates of response of >95% to 3 doses of standard vaccine, it is hard to argue that there is much effect at all due to genetic variation.

6. Would like to sub-analysis of biopsy proven cirrhotics only before concluding that cirrhosis does not make a difference. The overall data is suspect because carefully done prospective registration trials clearly show factors like DM and dialysis clearly impact seroprotection rates. The recent report by Jackson et al (Vaccine, 2019) shows that only 65.1% of subjects receiving Engerix-B developed a response after a full series.

7. Table 1: Confusing to read. I had to calculate the number of patients in each group. Would put N for each group

8. Disagree with comparisons using 2 dose recipients. Would focus on 3 dose only for reasons described above.

9. Data in Table 3 shows 3 doses matter. Would focus on that finding though it is not a surprise.

10. Would use multivariable statistical methods to assess impact of individual factors, not just univariate methods.

Reviewer #2: This manuscript summarizes a study of HBC vaccination among patients diagnosed with HCV. This is an important issue because treatment for HCV can reactivate HBV. The study evaluated patients of a Midwest health care system who were infected with HCV and vaccinated against HBV.

Major issue:

Throughout the paper it is unclear if HBV vaccination was initiated after HCV infection occurred.

Clarity on this point is needed throughout the paper.

Some results are not consistent internally.

Specific comments:

Note: because there are no line numbers I placed quotes to find section discussed.

In general, check grammar. In particular, check capitalized words.

Introduction

Estimates of HBV and HCV prevalence utilize old estimates. The improved estimates can be found at the WHO website:

https://www.who.int/hepatitis/publications/global-hepatitis-report2017/en/

Methods

“study, conducted” the comma should be removed

It would read better if the single sentence first paragraph was edited into two or more sentences.

Records that were available were reviewed. Whose records are not electronically available? About how many? Note: this information doesn’t need to be added to the manuscript if the number is extremely small (e.g., <5).

“Clinical information of the patients were collected” – consider beginning a new paragraph.

“Hepatitis B Surface Antibody (HBsAb)” add the word “test” or when blood drawn.

“Differences between continuous variables were evaluated at mean, using a T-Test for statistical significance.” The t-test compares the means. This sentence needs to be rewritten. Also, if you are using a t-test then it would help to provide the means and stardard deviations.

Results

“(n=1276) received various doses of HBV vaccination, 50% (n=646) of which were checked for HBsAb afterwards. Among those, 65% (n=418) developed HBsAb after vaccination (vaccine responders) and 35% (n=228) did not (vaccine non-responders).” This information is inconsistent with what is in Table 1. Specifically, in Table 1 there has 185 vaccine responders.

The other confusing issue is when patients received HBV vaccine. The methods suggests that only those who were vaccinated after HCV diagnosis are studied, however this isn’t clear throughout the paper.

A flow chart would greatly enhance the reader’s understanding of the data. The flow chart could also clarify if the HBV vaccines started before or after HCV diagnosis. Did you exclude patients who were vaccinated for HBV before HCV diagnosis? If so, providing this information on a flow chart would help the readers.

Tables in general would benefit from changing the percentages direction. The question of interest is: What proportion of those with risk factor X are responders?

Table 1 would also benefit from the addition of a line for patients who have been vaccinated according to the guidelines.

“median age at the time of the HBsAb check was 54 years in both of the groups (IQR: 47-59 and 46-50 for responders and non-responders respectively).” This can’t be correct for non-responders.

Comparison of patients who had two HBV vaccines lacks logic. The comparisons should be for patients who were vaccinated according to guidelines. You noted a difference between 2 and 3+ doses.

It would help if the analyses of risk factors are adjusted by age and sex. If age is put into categories then this adjusted analyses could still be conducted with PROC FREQ. This is particularly important for analysis of cirrhosis given that these individuals will be skewed toward older ages.

Table 3 is not helpful. It brings into the analysis patients who obtained one HBV vaccine.

Discussion

Recommend removing “largest” study given that there are likely larger studies.

The word “decreased” suggests a change. It may be more accurate to say “lower”.

In the discussion there is suggestion that patients were HBV vaccinated after HCV diagnosis. When vaccines occurred needs to be clear throughout the paper.

Overall, the discussion is well written. It is difficult to fully review the discussion section until the results are clarified.

6. PLOS authors have the option to publish the peer review history of their article (what does this mean?). If published, this will include your full peer review and any attached files.

Reviewer #1: No

Reviewer #2: No

---

## [Author Response · Author response to Decision Letter 0]

28 May 2020

The authors appreciate the reviewers’ thoughtful and insightful comments, as well as the time and effort spent. We have changed the manuscript accordingly, and revised several of the analyses per the reviewers’ requests. Below, we provide responses to each comment (please note that page and line refer to the marked-up version).

Reviewer #1:

1. Comment: “Page 5: The sentence regarding vaccination with Euvax B is very confusing. This vaccine is not licensed in the U.S. and the next sentence describes other vaccine products used.”

 Response: The comment about Euvax has been removed, page 6, line 127. 

2. Comment: “I disagree with defining vaccine non-responders as those that received at least 1 dose. While this may be the definition in an intent to treat trial, we know from prospective registration trials that the rate of response from those who received only 1 dose is quite low and 2 doses is suboptimal. In the original registration trials from Szumness, the post-vaccine response (defined as >10 mIU/ml) was around 30% and this went up to around 70% after the second dose. In prospective trials the testing was done at short time intervals while your testing was randomly done, also decreasing apparent outcomes. At the minimum your study should focus on those who completed a full series.”

 Response: We have updated our definition of non-responders to that of those receiving 3 or more doses of the vaccine. We have updated all the comparisons and tables accordingly. That has led us to remove some of the data (such as average number of vaccine doses in non-responders) and to re-calculate some analyses, however, for the most part, the results of the study remained the same. Table 2 now contains comparisons between all responders to non-responders (page 19), and table 3 (page 20) now contains comparisons between responders with 3 or more doses, to non-responders. Any previous data related to comparisons involving non-responders with less than 3 vaccine doses were removed from the manuscript. We also added results from Multivariate analysis, again, with non-responders being modified to only those who received 3 or more doses (page 9, lines 204-207). 

3. Comment “Page 6: Your definitions of cirrhosis are very non-specific. Why not apply FIB-4 which is imperfect but is at least standardized in HCV positive patients and call cirrhosis those with FIB-4 >3.25? Use of mildly increased bilirubin in the definition allows inclusion of those with Gilbert's gene polymorphisms.”

 Response: Our definition of cirrhosis included increased INR or increased bilirubin plus low platelet count AND cirrhotic looking liver (hence a combination of imaging and laboratory variables, which are recommended by the European Association for the Study of the Liver as referenced) or a combination of 3 laboratory parameters (two of coagulopathy and one of internal liver physiology). We believe there has been a confusion, and we have changed the wording for better clarification (page 6, lines 135-140). FIB4 and APRI scores have been reported to have a potential use for fibrosis assessment, but no international hepatology guideline supports them for cirrhosis diagnosis. 

4. Comment “Page 6: Only a subset of patients had testing for surface antibody after vaccination. How did these differ from the non-tested population?” 

 Response: Demographics showed that those not tested for HBsAb were 63% male compared to 70% male in those tested, 14% of those not tested had cirrhosis versus 25% of those tested (this yielded a significant difference, p=0.01). Mean ages were similar between both groups: 48 y/o (CI 47-50) vs tested 54 y/o (CI 53-54), p=0.001). We have added this information to the manuscript (Page 7, line 159-163). 

5. Comment “Page 8: Comments about genetic variation not germain to your results. Indeed, with general population rates of response of >95% to 3 doses of standard vaccine, it is hard to argue that there is much effect at all due to genetic variation.”

 Response: We have placed references that studied genetic variations (references 32-33). However, we understand the comment as a practical point. We have added a comment indicating the low likelihood of this playing a role (we believe it is worth mentioning at least). Page 10, lines 237-238. 

6. Comment “Would like to sub-analysis of biopsy proven cirrhotics only before concluding that cirrhosis does not make a difference. The overall data is suspect because carefully done prospective registration trials clearly show factors like DM and dialysis clearly impact seroprotection rates. The recent report by Jackson et al (Vaccine, 2019) shows that only 65.1% of subjects receiving Engerix-B developed a response after a full series.”

 Response: Our analysis is different than that quoted by the reviewer. The reviewer quotes a study in which 65% of cirrhotics respond to HBV vaccine. We mention that among our responders there were 24% cirrhotics, these are different ways of looking at data. We have looked at the response rate among cirrhotics that got 3 doses of the vaccine, and found a response rate of 77%. This was added to the body of the study (Page 8, lines 188-193). 

7. Table 1: Confusing to read. I had to calculate the number of patients in each group. Would put N for each group

 Response: Table 1 was modified as requested. Numbers of patients that received any number of doses was added (Table 1, Page 18).

8. Disagree with comparisons using 2 dose recipients. Would focus on 3 dose only for reasons described above.

 Response: We re-performed the analysis, assessing all responders vs non-responders with 3 or more doses (Table 2, Page 19) and Responders with 3 or more doses compared to non-responders with 3 or more doses (table 3, page 20). The “results” section, page 7, was re-written to indicate those changes and the numbers were updated. 

9. Data in Table 3 shows 3 doses matter. Would focus on that finding though it is not a surprise.

 Response: Similarly, as above, we have re-done the analysis with 3 doses and modified the table accordingly, as above. 

10. Would use multivariable statistical methods to assess impact of individual factors, not just univariate methods.

 Response: We have now included multivariate analysis to our study. On multivariate analysis we found that cirrhosis correlated with less reactivity compared to those without cirrhosis (OR 0.66, CI 0.44-0.94). In the multivariate analyses, alcohol abuse and diabetes did not show a significant correlation between HBsAb reactivity. This was added to the body of the text (Pages 9, lines 204-207). 

Reviewer #2: 

-Comment 1: “Throughout the paper it is unclear if HBV vaccination was initiated after HCV infection occurred. Clarity on this point is needed throughout the paper”.

 Response: As the reviewer can imagine it is certainly not possible to know when patients got infected with HCV, as the great majority of infections are subclinical, we can at best estimate based on risk factor timing. To provide an estimation for the reviewer, we did assessed the percentage of first HBV vaccinations in relation to the first positive test for HCV. From a small subset of individuals (255) in which we assessed this information, we found that 177 had the first vaccination before being tested for hepatitis C. It should be understood however that the great majority of individuals expressed a remoted risk factor for HCV infection, with likely linking of risk-factor to infection at least 1-2 decades before laboratory check. This was added to the methods section, Page 6, lines 129-132. 

-Comment 2: 

“Estimates of HBV and HCV prevalence utilize old estimates. The improved estimates can be found at the WHO website:

https://www.who.int/hepatitis/publications/global-hepatitis-report2017/en/”

 Response: We have added new estimates per the reviewer’s request (Page 4, lines 76-77). 

-Comment 3: “study, conducted” the comma should be removed

 Response: comma removed (Page 5, line 104). 

-Comment 4: “It would read better if the single sentence first paragraph was edited into two or more sentences.”

 Response: The paragraph was edited into 2 sentences (Page 5, lines 104-107) 

-Comment 5:” Records that were available were reviewed. Whose records are not electronically available? About how many? Note: this information doesn’t need to be added to the manuscript if the number is extremely small (e.g., <5)”.

 Response: Quote “all available” just means that all the records that we had were reviewed. We changed the paragraph per reviewers request (Page 5, line 108).

-Comment 6: “Clinical information of the patients were collected” – consider beginning a new paragraph.

 Response: The sentence was changed to a new paragraph (Page 5, line 113). 

-Comment 7: “Hepatitis B Surface Antibody (HBsAb)” add the word “test” or when blood drawn.

 Response: We added “test” following HBsAb throughout the study. 

-Comment 8: “Differences between continuous variables were evaluated at mean, using a T-Test for statistical significance.” The t-test compares the means. This sentence needs to be rewritten. Also, if you are using a t-test then it would help to provide the means and standard deviations.

 Response: We have changed the wording of the phrase after talking to our Statistitian to: “Statistical analyses were performed using SAS version 9.4. Differences between continuous variables were evaluated at the means using a T-Test. Categorical variables were evaluated using a Chi-Square test. The multivariate regression was modeled as a logistic regression.” This is reflected on Page 6, lines 147-149. 

-Comment 9: “(n=1276) received various doses of HBV vaccination, 50% (n=646) of which were checked for HBsAb afterwards. Among those, 65% (n=418) developed HBsAb after vaccination (vaccine responders) and 35% (n=228) did not (vaccine non-responders).” This information is inconsistent with what is in Table 1. Specifically, in Table 1 there has 185 vaccine responders.

 Response: The 185 patients noted as responders in table 1 are the responders with known number and timing of doses. The 418 responders noted in the text are all patients with documented vaccination in the EMR, but not all had available timing and number of doses.

Only the 185 patients were included in analyses pertaining to number of doses, however the remainder of patients were included in analyses regarding clinical and virus related variables, as these were available. Notably, we also re-did all the analyses for those with known 3 or more doses in the responder arm, compared to non-responders (table 3, page 20) and that yielded similar results when compared to the analyses in table 2, page 19. This was also clarified in the results section of the study, page 8, lines 200-202. 

-Comment 10: ‘The other confusing issue is when patients received HBV vaccine. The methods suggests: that only those who were vaccinated after HCV diagnosis are studied, however this isn’t clear throughout the paper.” 

 Response: This point was address above. We have indicated the number of individuals that received vaccination before and after testing positive to clarify this point, page 6, lines 129-132.

-Comment 11: “-A flow chart would greatly enhance the reader’s understanding of the data. The flow chart could also clarify if the HBV vaccines started before or after HCV diagnosis. Did you exclude patients who were vaccinated for HBV before HCV diagnosis? If so, providing this information on a flow chart would help the readers.”

 Response: A flow chart was added (Page 17, figure 1). 

-Comment 12: “Tables in general would benefit from changing the percentages direction. The question of interest is: What proportion of those with risk factor X are responders?”

Response: We appreciate the reviewers comment. However, our study is looking at which factors are associated with no response, which we believe is reflected in our tables. Changing the positioning of the tables would require changing also the analyses and reflective p values and we believe that would be a total different manuscript than the one we are submitting. We have however, considerably modified the tables based on the re-analysis. 

-Comment 13: “Table 1 would also benefit from the addition of a line for patients who have been vaccinated according to the guidelines.”

 Response: All patients with 3 doses were vaccinated per the guidelines. Those in the study who had interrupted timings restarted the 3 dose sequence and ended up receiving 4 or more doses. 

-Comment 14 “median age at the time of the HBsAb check was 54 years in both of the groups (IQR: 47-59 and 46-50 for responders and non-responders respectively).” This can’t be correct for non-responders.

 Response: The age analysis was redone to identify non-responders as those who did not respond after getting at least 3 doses. Means have now changed for non-responders (Page 8, lines 177-179).

-Comment 15: “Comparison of patients who had two HBV vaccines lacks logic. The comparisons should be for patients who were vaccinated according to guidelines. You noted a difference between 2 and 3+ doses.” 

 Response: Comparisons between patients with 2 doses were removed. Now the comparison is between patients who received 3 or more doses (Page 20, table 3). 

-Comment 16: “It would help if the analyses of risk factors are adjusted by age and sex. If age is put into categories then this adjusted analyses could still be conducted with PROC FREQ. This is particularly important for analysis of cirrhosis given that these individuals will be skewed toward older ages.” 

 Response: We have performed and added a multivariate analysis and included age per reviewer’s request (Pages 9, lines 204-207). 

-Comment 17: “Table 3 is not helpful. It brings into the analysis patients who obtained one HBV vaccine.”

 Response: Table 3 now modified and analysis was re-done to include patients who received 3 or more vaccines only (Page 20, table 3).

-Comment 18: “Recommend removing “largest” study given that there are likely larger studies.”

 Response: Through our literature review, we have not encountered a study of a similar size, aside from the noted meta-analysis. However, the phrase was substituted with “one of the largest studies”, per the reviewer’s request (Page 9, line 210).

-Comment 19: “The word “decreased” suggests a change. It may be more accurate to say “lower”.

Response: The word “Decreased” was switched to lower, when pertaining to the response to the vaccine, throughout the study. 

-Comment 20 “In the discussion there is suggestion that patients were HBV vaccinated after HCV diagnosis. When vaccines occurred needs to be clear throughout the paper”

 Response: We have added a clarification on the percentage of patients that received vaccination before or after diagnosis. We would like to emphasize that this really does not correlate with “hepatitis C acquisition” as that is not possible to define, since most infections are subclinical. Clarification was added to the “methods”, page 6, lines 129-132. 

Additional comments: 

-A relatively conservative estimate of HBV vaccine protectiveness was used. What if that cutoff were increased to 100?

 Response: Our cut off (a titer of 12 mIU/mL) is above the recommended cut off of 10 mIU/mL, deemed to be protective against hepatitis B (West DJ, Calandra GB. Vaccine induced immunologic memory for hepatitis B surface antigen: implications for policy on booster vaccination. Vaccine. 1996;14(11):1019‐1027.), which is also the cut off recommended by the CDC. 

-Were the sociodemographic or clinical characteristics of those that were checked for anti-HBs different from those that were not checked? (Regardless of the findings)

Response: We assessed the characteristics of those who were tested to document immunity and those who were not, patients who were not tested were younger, both groups were predominantly male, this data was added (Page 7, Lines 159-163).

 -What was the HIV status of the individuals in the responder versus non-responder groups?

 Response: Because of the low number of individuals with HIV (<2%) in the population, these individuals were removed from the study. 

- How were alcohol and intravenous drug use defined? Self-report? Ever / never?

Response: History of alcohol abuse, was obtained from the chart, by assessing for documentation of use compatible with abuse or for a diagnosis of alcohol abuse, related to recurrent alcohol related presentations and positive toxicology in those who did not self-report heavy use. To ensure the patients had active alcohol abuse at the time of vaccination, only patients who were noted to have ongoing abuse in the hepatology clinic were defined as abusers for the purpose of the study. 

IV drug use was obtained from EMR charts and problem lists that could have been a consequence of previous positive toxicology or relied on self-report. Patients were considered to have IV drug abuse if the evaluating provider noted a history of active or recent drug use on presentation to the hepatology clinic.

---

## [Decision Letter · Decision Letter 1]

23 Jun 2020

PONE-D-20-06696R1

Response to hepatitis B virus vaccination in individuals with chronic hepatitis C virus infection

PLOS ONE

Dear Dr. Ashhab,

Thank you for submitting your manuscript to PLOS ONE. After careful consideration, we feel that it has merit but does not fully meet PLOS ONE’s publication criteria as it currently stands. Therefore, we invite you to submit a revised version of the manuscript that addresses the points raised during the review process.

Reviewer 1 remains unconvinced of the methods and findings.  Please address the comments from reviewer 1 thoroughly in a revised manuscript.

We look forward to receiving your revised manuscript.

Kind regards,

Jason Blackard, PhD

Academic Editor

PLOS ONE

Reviewers' comments:

Reviewer's Responses to Questions

**Comments to the Author**

1. If the authors have adequately addressed your comments raised in a previous round of review and you feel that this manuscript is now acceptable for publication, you may indicate that here to bypass the “Comments to the Author” section, enter your conflict of interest statement in the “Confidential to Editor” section, and submit your "Accept" recommendation.

Reviewer #1: (No Response)

Reviewer #2: All comments have been addressed

2. Is the manuscript technically sound, and do the data support the conclusions?

Reviewer #1: No

Reviewer #2: Yes

3. Has the statistical analysis been performed appropriately and rigorously? 

Reviewer #1: No

Reviewer #2: Yes

4. Have the authors made all data underlying the findings in their manuscript fully available?

Reviewer #1: No

Reviewer #2: No

5. Is the manuscript presented in an intelligible fashion and written in standard English?

Reviewer #1: Yes

Reviewer #2: Yes

6. Review Comments to the Author

Reviewer #1: This revised manuscript provides a retrospective analysis of the HBV vaccine response rates to a large number of HCV infected persons. The authors primary conclusion is that that cirrhosis is a key factor in failure to respond.

Specific Comments

1. Abstract: The authors conclude that HCV infection impairs HBV vaccine response with cirrhosis being the only identifiable risk factor. This conclusion is not supported by the data presented. There is no evidence that HCV specifically impairs HBV vaccine response. Indeed, the statement actually says that cirrhosis is the factor that impairs response (among those with HCV). This distinction is important.

2. Page 7: I remain confused about the numbers and calculations. The rate of response in those that received 3 doses was 42%. Then the authors note that 77% of those that received 3 or more doses had a response but in parentheses it says (32/139). This is a 23% response rate.

3. The overall response rate in non-cirrhotic patients who completed 3 or more doses is not explicitly stated. However, the overall rate of response among those who received 3 or more doses was 79.5%.

4. In univariate analysis there was no difference in response related to cirrhosis with the proportions of cirrhotics among responders and non-responders essentially equal (p= 0.6). To then select a subset of characteristics and find significance for cirrhosis is a bit of statistical gamesmanship.

5. Page 7: After stating that 418/525 (79.6%) have a vaccine response, the next sentence says "Out of all who receive 3 or more doses only 50% develop HBsAb.

Reviewer #2: All comments were sufficiently addressed.

The data is not currently available. However, the authors state they will make the data available upon publication. This meets the goal of the requirement.

7. PLOS authors have the option to publish the peer review history of their article (what does this mean?). If published, this will include your full peer review and any attached files.

Reviewer #1: No

Reviewer #2: No

---

## [Author Response · Author response to Decision Letter 1]

7 Jul 2020

The authors appreciate the reviewer’s thoughtful and insightful comments, as well as the time and effort spent. 

1. Abstract: The authors conclude that HCV infection impairs HBV vaccine response with cirrhosis being the only identifiable risk factor. This conclusion is not supported by the data presented. There is no evidence that HCV specifically impairs HBV vaccine response. Indeed, the statement actually says that cirrhosis is the factor that impairs response (among those with HCV). This distinction is important.

Response: We appreciate the reviewer’s comment. Our data shows a 79% HBV vaccine response, that is much lower than the >90% reported in the general population in multiple studies, and a 50% response rate in the population who received 3 or more doses. We have changed the comment in the abstract to clarify (page 3, line 66). To further clarify this to the readers, we have added the response rates in non-cirrhotic HCV patients in our cohort, which is also well below the rate reported in the literature, to the results section (page 8, lines 188-191). 

2. Page 7: I remain confused about the numbers and calculations. The rate of response in those that received 3 doses was 42%. Then the authors note that 77% of those that received 3 or more doses had a response but in parentheses it says (32/139). This is a 23% response rate.

Response: We apologize for the confusion regarding the second portion of the comment, we made a typo (32 were non responders, but 107 were, accounting for a response rate of 77% as indicated). We have fixed this issue in the manuscript (page 8, line 187). We have to emphasize however, that as indicated in that line, this is the response rate in cirrhotics who received 3 or more doses. When assessing the entire population who received 3 or more doses, is detailed in table 3, page 20, response rate is 50% (equal number of patients in each arm). 

* Clarification: The “42%” noted in table 1, is not the response rate, but rather, the proportion of responders, who sero-converted following 3 doses of the vaccine, as is indicated in the column title. 

3. The overall response rate in non-cirrhotic patients who completed 3 or more doses is not explicitly stated. However, the overall rate of response among those who received 3 or more doses was 79.5%.

 Response: Thank you for the comment. As we indicated above, we have added the lines 188-191, in page 8, to clarify the response rate in non-cirrhotics only, which as indicated above, is significantly lower than the response rate in the general population. 

4. In univariate analysis there was no difference in response related to cirrhosis with the proportions of cirrhotics among responders and non-responders essentially equal (p= 0.6). To then select a subset of characteristics and find significance for cirrhosis is a bit of statistical gamesmanship.

 Response: We appreciate the reviewer’s comment. However, we carefully chose the characteristics (Age, gender, alcohol abuse, and DM) as each of these has been reported to be associated with decreased HBV vaccine response in the literature. We have added a comment to clarify this (Page 9, line 203). Seeing as the proportion of Cirrhosis was higher in the non-responders’ arm, but no statistically, a concern was that those characteristics may have been confounders that limited the univariate analysis. A multivariate analysis accounting for possible confounders was also suggested by reviewer 2 at the time of the initial submission. 

5. Page 7: After stating that 418/525 (79.6%) have a vaccine response, the next sentence says "Out of all who receive 3 or more doses only 50% develop HBsAb.

Response: Our initial statement says “various doses” indicating that we are including patients that responded even with 2 doses of the vaccine. The statement includes those that received all 3 doses. We have changed “various” to “any number” of HBV vaccine doses to make that more clear (page 7, line 161-162).

---

## [Decision Letter · Decision Letter 2]

27 Jul 2020

Response to hepatitis B virus vaccination in individuals with chronic hepatitis C virus infection

PONE-D-20-06696R2

Dear Dr. Ashhab,

We’re pleased to inform you that your manuscript has been judged scientifically suitable for publication and will be formally accepted for publication once it meets all outstanding technical requirements.

Kind regards,

Jason Blackard, PhD

Academic Editor

PLOS ONE

Additional Editor Comments (optional):

None

Reviewers' comments:

Reviewer's Responses to Questions

**Comments to the Author**

1. If the authors have adequately addressed your comments raised in a previous round of review and you feel that this manuscript is now acceptable for publication, you may indicate that here to bypass the “Comments to the Author” section, enter your conflict of interest statement in the “Confidential to Editor” section, and submit your "Accept" recommendation.

Reviewer #1: All comments have been addressed

Reviewer #2: All comments have been addressed

2. Is the manuscript technically sound, and do the data support the conclusions?

Reviewer #1: Yes

Reviewer #2: Yes

3. Has the statistical analysis been performed appropriately and rigorously? 

Reviewer #1: Yes

Reviewer #2: (No Response)

4. Have the authors made all data underlying the findings in their manuscript fully available?

Reviewer #1: Yes

Reviewer #2: No

5. Is the manuscript presented in an intelligible fashion and written in standard English?

Reviewer #1: Yes

Reviewer #2: Yes

6. Review Comments to the Author

Reviewer #1: Revised manuscript addresses key issues.

Might want to include 95% C.I. of overall response rate among HCV positive (78-84%) which does indeed support rate being lower than non-HCV healthy population.

Reviewer #2: The authors will make data available upon publication. This is acceptable.

Only style changes I recommended previously were not made. This is ok.

7. PLOS authors have the option to publish the peer review history of their article (what does this mean?). If published, this will include your full peer review and any attached files.

Reviewer #1: No

Reviewer #2: No

---

## [Editor Report · Acceptance letter]

13 Aug 2020

PONE-D-20-06696R2 

Response to hepatitis B virus vaccination in individuals with chronic hepatitis C virus infection 

Dear Dr. Ashhab:

I'm pleased to inform you that your manuscript has been deemed suitable for publication in PLOS ONE. Congratulations! Your manuscript is now with our production department. 

Kind regards, 

on behalf of

Dr. Jason Blackard 

Academic Editor

PLOS ONE